

# Pre-touch reaction is preferred over post-touch reaction in interaction with displayed agent

Masahiro Shiomi

Interaction Science Laboratories, Advanced Telecommunications Research Institute International, Seika-cho, Kyoto, Japan

## ABSTRACT

A pre-touch reaction, which is a response before a physical contact, is an essential factor for natural human-agent interaction. Although numerous studies have investigated the effectiveness of pre-touch reaction design for virtual agents in virtual reality (VR) environments and robots in physical environments, one area remains underexplored: displayed agents, *i.e.*, on-screen computer graphics agents. To design an appropriate pre-touch reaction for such a displayed agent, this article focused on the display's physical boundary as a criterion for the pre-touch reaction of the agent. This article developed a displayed agent system that can detect both the touch events on the screen and the pre-touch behaviors of the interacting people around the display. This study examined the effectiveness of the pre-touch reactions of the displayed agent by the developed system in experiments with human participants. The findings revealed that people significantly preferred pre-touch reactions over post-touch reactions in the context of perceived feelings.

## INTRODUCTION

Human beings typically respond before physical contact occurs when they perceive an intention of possibly being touched by others. Such interaction, known as a pre-touch reaction, is a fundamental aspect of human-agent interaction design. For instance, numerous researchers have explored the optimal distance for pre-touch reactions in human-agent interaction, including scenarios where robots act as physical agents (*Cuello Mejía et al., 2021a*; *Kimoto et al., 2023*; *Cuello Mejía et al., 2023*; *Shiomi et al., 2018*). These studies have consistently shown that pre-touch reactions are preferred more than post-touch reactions (*i.e.*, reactions after being touched) in human-robot interaction. They have also provided valuable insights for designing pre-touch reaction behaviors for interactive agents to contribute to the creation of acceptable and human-like interactions with people.

Although such an interaction guideline exists, pre-touch interaction design is less focused on a displayed agent, *i.e.*, on-screen computer graphics agents (Fig. 1). For example, past studies conducted several physical and spatial interactions with such displayed agents as f-formation (*Hedayati, Szafir & Kennedy, 2020*) for supporting children with autism (*Bernardini, Porayska-Pomsta & Smith, 2014*), haptic interaction with a mechanical hand device (*Kumagai & Nonomura, 2023*; *Onishi et al., 2024*), an

Corresponding author
Masahiro Shiomi, m-shiomi@atr.jp

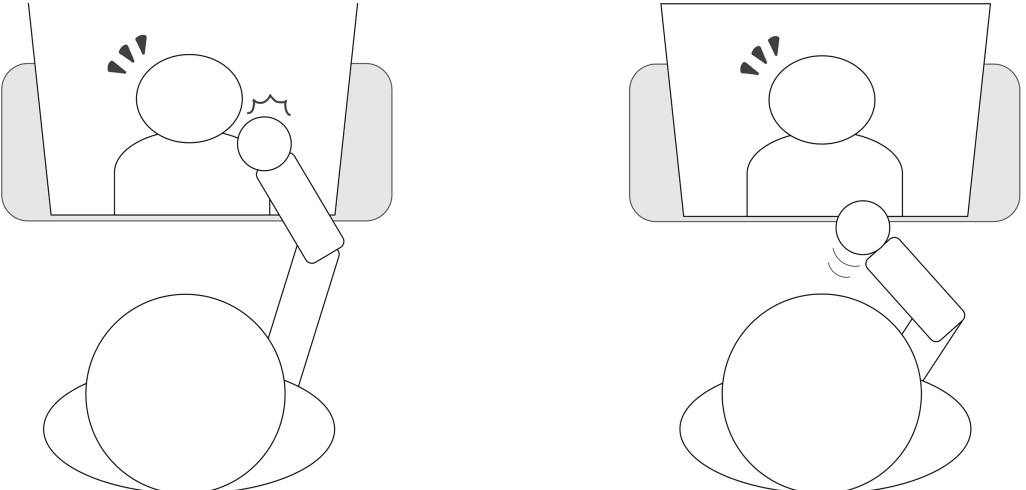

**Figure 1 Illustration of reactions after touch (left) and before touch (right) with displayed agents.**

illusion of touch interaction (*Mercado, Bailly & Pelachaud, 2016*), and an analysis of interactions with multiple people (*Peltonen et al., 2008*). These studies identified the utility of displayed agents in the context of physical and spatial interaction, although they less focused on pre-touch interaction.

There is one difficulty in designing pre-touch interaction for such displayed agents: reaction distance criteria. One notable difference between the displayed agents and robots/agents is the existence of a display that will have an influence on physical interaction. Unlike robots in physical environments and agents in virtual reality (VR) environments, it is unclear whether the criterion for the pre-touch reaction distance to which the display agents should respond is the distance from the face on the displayed image or to the display's physical boundary. If the distance between the displayed face and the touching hand is essential for pre-touch reactions, achieving pre-touch reactions will be complicated when the displayed agent's position is far from the display surface. On the other hand, if the display's surface is the criterion for pre-touch reactions, displayed agents can easily create pre-touch reactions before the display is touched.

This study addresses these issues for pre-touch reactions for displayed agents by developing a displayed agent system that consists of a touchscreen to detect touch interaction on the display's surface and a depth sensor to detect the hand positions around its surface. Moreover, through an experiment with participants, this article investigates whether pre-touch reactions are preferred over post-touch reactions in interaction with displayed agents. The structure of this article is as follows. "Related Work" provides a detailed description of related research and outlines the position of this study. "Materials and Methods" explains the developed displayed agent system used in the experiment and its settings. "Results" presents the experimental results. "Discussion" discusses the implications arising from the results related to the perceptions toward pre-touch reactions. Finally, "Conclusions" summarizes the article.

## RELATED WORK

### Interaction with displayed agents

Past studies have mainly focused on conversational interaction with displayed agents. For example, *DeVault et al. (2014)* developed a displayed agent for conducting interviews about healthcare decision support, and their colleagues conducted a series of experiments using virtual agents to support people's mental health from various perspectives (*Gratch et al., 2014*; *Hartanto et al., 2015*; *Rizzo et al., 2016*). Displayed agents have also been used for applying cognitive behavior therapy (CBT) to people to support ameliorating automatic thoughts (*Kimani et al., 2019*; *Shidara et al., 2022*). To achieve natural conversational interaction with such agents, researchers investigated appropriate multi-modal interaction designs, including gestures and gaze through experiments where people interacted with displayed agents (*Andrist et al., 2012*; *Babu et al., 2006*; *Potdevin, Clavel & Sabouret, 2021*). Moreover, a past study has focused on spatial interaction with such displayed agents, *e.g.*, scenes where people encountered the displayed agents, and reported different interaction distances due to the people's impressions toward the agents (*Cafaro, Vilhjálmsson & Bickmore, 2016*). From another perspective, a past study investigated the visualization effects of remote touch interaction in distant locations between teleoperated avatars and users, and reported that the approach increase the perceptions of being touched by the operators (*Kimoto & Shiomi, 2024*). They also suggested changing reaction behaviors depending on the places that are actually touched for refusing excessive or "bad" touches.

These studies gathered rich knowledge for designing conversational displayed agents and their interaction styles for natural conversations with people. Unfortunately, they mainly focused on conversational interaction, instead of pre-touch interaction, which is this study's focus.

### Pre-touch reaction behaviors design for agents

Past studies concluded that a robot's reactions before being touched are essential to provide a human-like and natural impression to people who are preparing to touch them. For example, *Shiomi et al. (2018)* modeled the pre-touch reaction distance around the face based on the data collection with people and investigated the effects of the pre-touch reaction distance model with a social robot. *Cuello Mejía et al. (2021a)* also modeled pre-touch reaction distances not only around faces but also upper bodies and implemented a pre-touch reaction distance for human-like pre-touch reactions in a social robot. The pre-touch reaction is also critical in touch interaction in VR environments. *Cuello Mejía et al. (2021b)* compared the pre-touch reaction distances between physical and VR environments and concluded that these pre-touch reaction distances are not significantly different between these environments. To achieve a more human-like pre-touch reaction before touching, *Cuello Mejía et al. (2023)* investigated what kinds of gaze behaviors are perceived as more natural as pre-touch reaction behaviors and reported that complex gaze behaviors (*e.g.*, looking at both a touching hand and a toucher's face) are perceived as more natural than simple gaze behaviors. From another perspective, *Kimoto et al. (2023)*

investigated the relationships between the appearances of virtual agents in both touchers and receivers and reported that robot-like appearances decreased pre-touch reaction distances.

These studies have contributed knowledge for designing pre-touch reaction behaviors and appropriate pre-touch reaction distances to provide human-like impressions. However, unfortunately, their focus is limited to social robots and virtual agents, not displayed agents.

### Post-touch reaction behavior designs for agents

Post-touch reaction behaviors are also essential to achieve natural human-agent interaction. For example, *Okuda, Takahashi & Tsuichihara (2022)* reported that people who touch a social robot showed positive feelings when it appropriately responded to their social touch. *Lehmann et al. (2022)* investigated the appropriate reactions of robots toward people's touch and reported that the leaning-back behaviors of robots were positively evaluated. *Burns, Ojo & Kuchenbecker (2023)* concluded that the emotional reactions of a child-sized zoomorphic robot toward touch interaction were positively evaluated. From another perspective, *Kimoto & Shiomi (2024)* investigated the perceived feelings of being touched when teleoperated avatars are touched at remote places. The visualization of post-touch reactions increased feelings of being touched and discomfort toward it.

These studies have supplied knowledge for designing appropriate post-touch reaction behaviors and broadly investigated the perceived impressions toward such reaction behaviors. Unfortunately, their focus is often limited to post-touch interactions, not on pre-touch interaction.

## MATERIALS AND METHODS

### System

A displayed agent system that can react to both pre- and post-touch behaviors from interacting partners is developed for this study. Figure 2 shows an overview of the developed system that consists of a touchscreen display to detect touch behaviors, a depth sensor to detect pre-touch behaviors, and a displayed agent control system.

The touchscreen display (DELL P2418HT) detects the touched position on the screen when a person is touched. In this study, a depth sensor (Ultraleap 3Di) is installed on the front of the display that detects the hand position around the touchscreen between 0 to 50 cm from the display's front. This sensor enables the system to calculate the distance between the display surfaces (in the middle of the displayed agent's face) and the detected hands of the participants. Both systems can send the touched events and the distances between the display surfaces and the detected hand by a TCP/IP connection to the display agent control system (Unity-based software), which manages the displayed agent's motions. VRoidStudio (*Isozaki et al., 2021*) is used in this study to prepare a displayed agent. The control system plays idle motions where the agent looks down and breathes; when the system receives touched events or the distance between the display and the people's hand falls below thresholds, it plays a reactive behavior that causes the agent to look to the front.

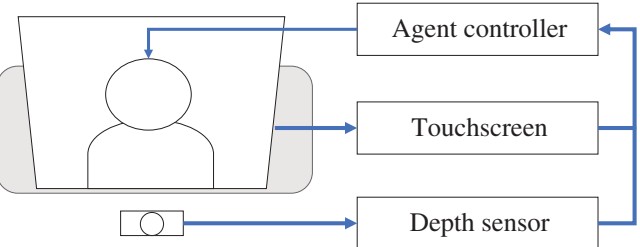

**Figure 2 Overview of the developed displayed agent system.**

After releasing the touched events or the distance between the display and the people's hand exceeds the thresholds, the system again plays idle motions for the agent.

## Hypothesis and prediction

The aim of this study is to investigate the perceived feelings of participants toward the displayed agent's pre- and post-touch reaction behaviors. Past studies suggested that a pre-touch reaction of 20 cm from the faces of the agents will be more positively evaluated than a post-touch reaction for both robots in physical environments and virtual agents in VR environments (*Cuello Mejía et al., 2021b*; *Shiomi et al., 2018*). Such pre-touch reactions might increase the human likeness of these agents, which would correlate with positive evaluations. Similar to these contexts, it is expected that the pre-touch reactions of the displayed agents would increase the perceived impressions of anthropomorphism and human likeness. Past studies reported that anthropomorphism affects the trust and acceptance of the interacting agents (*Natarajan & Gombolay, 2020*; *Roesler, Manzey & Onnasch, 2021*), which might also positively affect the perceived feelings. Thus, the participants might prefer a pre-touch reaction over a post-touch reaction even though the interaction target is a displayed agent.

Perhaps the display's physical boundary is appropriate for regarding the origin to investigate a pre-touch reaction distance rather than the face of the projected image. In other words, even though the position of the displayed agent's face is projected more than 20 cm back, the appropriate pre-touch reaction distance is 20 cm from the display. If this assumption is correct, people will prefer a pre-touch reaction 20 cm from the display regardless of the apparent distance from the displayed agent.

Based on these considerations, the following prediction was made: the participants will prefer a pre-touch reaction compared to a post-touch reaction in interactions with a displayed agent. To validate the prediction, the experiment in this study followed the experimental tasks of past studies, which investigated the pre-touch reaction distance for social robots and virtual agents (*Cuello Mejía et al., 2021b*; *Shiomi et al., 2018*). In this task, the participants moved their hands to touch the displayed agent's face until it reacted.

## Conditions

The experiment in this study investigated two factors: (1) touch and (2) viewpoint. Both were treated as within-subject factors, resulting in each participant experiencing four conditions that combined two levels of reactions toward touch with two distances.

(1) Touch (two levels: pre-touch reaction/touch reaction, within-subject): This study investigated the effects of using/not using the pre-touch reaction in the being-touched situation. In the pre-touch level, the displayed agent reacts before being touched (Fig. 3, right columns); the threshold of the pre-touch reaction distance is 20 cm, following past studies (Cuello Mejía et al., 2021b; Shiomi et al., 2018). In the touch level, the displayed agent reacts after touch events are detected by the touchscreen (Fig. 3, left columns).

(2) Viewpoint (two levels: the viewpoint is 20/45 cm far from the displayed agent's face, within-subject): This study investigated the effects of different viewpoints on the displayed agents. The developed system used a 20 cm as the pre-touch threshold distance based on past studies (Cuello Mejía et al., 2021b; Shiomi et al., 2018), which concluded that a 20 cm distance from the face is appropriate as a pre-touch reaction distance (Fig. 3, top row). The developed system used a 45 cm as the intimate distance for people, as previously defined by a proxemics study (Hall, 1966) (Fig. 3, bottom row). The reason for preparing this factor is to investigate whether participants preferred pre-touch reactions regardless of the apparent distance from the displayed agent. Moreover, examining the 45 cm setting might reinforce the evidence that the physical boundary of the display is more important as a criterion for the pre-touch distance than the apparent distance.

## Procedure

The participants first received explanations about the experiment and filled in a consent form. Then, they sat in front of the display in an experiment room and looked at the displayed agent system. The experimenter asked them to move their dominant hands toward the displayed agent's face, an action they did until the displayed agent reacted; finally, they filled out questionnaires. They repeated this procedure four times, combining two levels of touch and viewpoint factors. The combinations of factors and topics and their presentation order were counterbalanced to avoid order effects. In this study, only one displayed agent type (male) is prepared because past studies reported that the gender of the agents did not have any significant effects on the pre-touch reaction distance (Cuello Mejía et al., 2021b). All the procedures were approved by the Advanced Telecommunications Research Institute International Review Board Ethics Committee (501-4).

## Participants

Thirty participants joined the study: 15 men and 15 women, all of whom were native Japanese speakers (mean age = 39.43 years, S.D was 11.56). They were recruited through a local commercial recruiting company in Japan; they were then paid for participation. They had diverse backgrounds, such as students and business people. The analysis with the

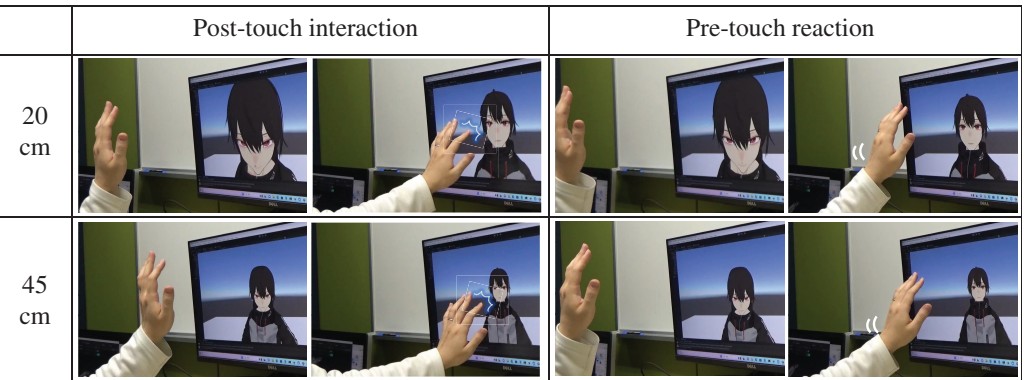

**Figure 3 Pre- and post-touch reactions of the displayed agent in each condition.** Images made using VRoidStudio.                                    

G*power (*Faul et al., 2007*) (a medium effect size of 0.25, a power of 0.8, and an α level of 0.05) showed that the sample size was 24, which indicated that the number of participants in this study was enough. Figure 3 shows interaction scenes between a participant and the display agent in each condition. After experiencing the (pre-) touch interaction with the agents, they again filled out questionnaires (see the next subsection) in the same room.

### Measurement

To evaluate the perceptions of the displayed agent's reactions, the following questionnaire items are used by referring to related studies: *anthropomorphism* (five items: fake-natural, machinelike-humanlike, unconscious-conscious, artificial-lifelike, and moving rigidly-moving elegantly) and *likeability* (five items: dislike-like, unfriendly-friendly, unkind-kind, unpleasant-pleasant, and awful-nice) from the Godspeed scale (*Bartneck et al., 2009*), which is commonly used in human-robot and human-agent interaction studies to investigate the perceived impressions during interaction. Some of them are excluded because they are not applicable in the experiment setting (*e.g.*, perceived safety). The items were evaluated in a seven-point response format where 1 was the most negative and 7 was the most positive, the average values of each item for the analysis. *Cronbach*'s *(1951)* alpha (*Cronbach, 1951*) values of *anthropomorphism* (0.938) and *likeability* (0.935) exceeded the required criterion.

Moreover, the three kinds of single items are used to investigate the perceived impressions toward pre-touch reaction based on the related studies (*Cuello Mejía et al., 2021b*; *Shiomi et al., 2018*): *human likeness* ("I think that the robot's reaction distance is human-like"), *naturalness* ("I think that the robot's reaction distance is natural"), and *total feeling* ("I have a good impression of the robot overall"). The items were also evaluated in a seven-point response format where 1 is the most negative, and 7 is the most positive.

## RESULTS

Figure 4 shows the questionnaire results. This analysis conducted a two-way repeated measure analysis of variance (ANOVA) for the *touch* and *viewpoint* factors to investigate the F-values ($F$), the $p$-values ($p$), and the effect size (partial eta squared, $\eta_p^2$).

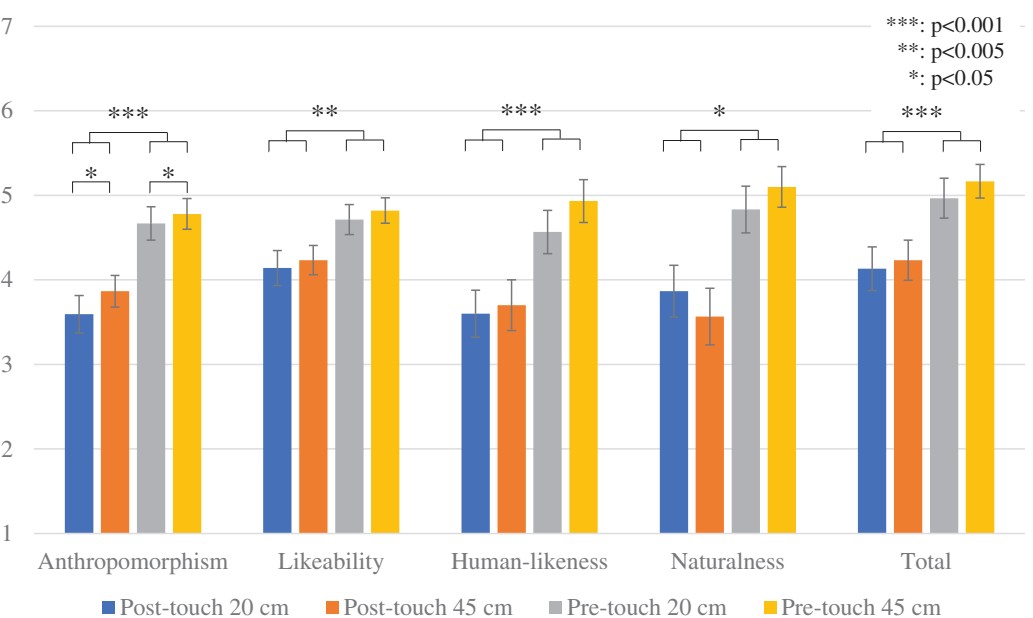

**Figure 4 Results of questionnaire items: error bars represent standard error.** Each graph indicates the average questionnaire results and standard error.

For *anthropomorphism*, the analysis found significant differences in the *touch* factor ($F$ $(1.29) = 24.229$, $p < 0.001$, $\eta_p^2 = 0.455$) and the *viewpoint* factor ($F (1.29) = 4.453$, $p = 0.044$, $\eta_p^2 = 0.133$). The analysis did not find a significant difference in the interaction effect ($F$ $(1.29) = 0.314$, $p = 0.580$, $\eta_p^2 = 0.011$).

For *likeability*, the analysis found a significant difference in the *touch* factor ($F (1.29) = 10.678$, $p = 0.003$, $\eta_p^2 = 0.269$). The analysis did not find significant differences in the *viewpoint* factor ($F (1.29) = 1.480$, $p = 0.234$, $\eta_p^2 = 0.049$) or in the interaction effect ($F$ $(1.29) = 0.003$, $p = 0.957$, $\eta_p^2 = 0.001$).

For *human likeness*, the analysis found a significant difference in the *touch* factor ($F$ $(1.29) = 15.435$, $p < 0.001$, $\eta_p^2 = 0.347$). The analysis did not find significant differences in the *viewpoint* factor ($F (1.29) = 2.895$, $p = 0.095$, $\eta_p^2 = 0.093$) or in the interaction effect ($F$ $(1.29) = 0.369$, $p = 0.549$, $\eta_p^2 = 0.013$).

For *naturalness*, the analysis found a significant difference in the *touch* factor ($F (1.29) = 9.481$, $p = 0.005$, $\eta_p^2 = 0.246$). The analysis did not find significant differences in the *viewpoint* factor ($F (1.29) = 0.009$, $p = 0.924$, $\eta_p^2 = 0.001$) or in the interaction effect ($F$ $(1.29) = 1.524$, $p = 0.227$, $\eta_p^2 = 0.050$).

For *total feeling*, the analysis found a significant difference in the *touch* factor ($F (1.29) = 13.758$, $p < 0.001$, $\eta_p^2 = 0.322$). The analysis did not find significant differences in the *viewpoint* factor ($F (1.29) = 1.943$, $p = 0.174$, $\eta_p^2 = 0.063$) or in the interaction effect ($F$ $(1.29) = 0.148$, $p = 0.703$, $\eta_p^2 = 0.005$).

These results suggest that a pre-touch reaction is preferred over post-touch reactions in interactions with displayed agents. Therefore, these results support the prediction. Note that the *viewpoint* factor did not show any significant effects except for the *anthropomorphism* scale.

## DISCUSSION

### Implications

The experiment results indicated that a pre-touch reaction by the displayed agents is preferred, similar to robots in physical environments and virtual agents in VR environments. This experiment's results provide several implications.

First, the developed system implemented a simple reaction behavior to investigate its effectiveness for a displayed agent. However, a past study reported complex gaze behaviors, such as looking at a hand being touched and the toucher's face, were more preferred (*Cuello Mejía et al., 2023*). To achieve such reaction behaviors, implementing a face detection system toward displayed agents would be useful using such common image-processing libraries as OpenCV (*Bradski, 2000*), YOLO (*Chen et al., 2021*) and OpenPose (*Cao et al., 2017*).

This study only investigated the pre-touch reaction effects where the displayed agent is aware of the touching behaviors before being touching, *i.e.*, looking to the front. If the displayed agent's direction were different, *e.g.*, having its back to the people, the pre-touch reaction of the displayed agent might appear strange toward them; such post-touched reactions would be more natural. Therefore, an appropriate pre-touch reaction behavior design will change based on the displayed agent's situation. Gaze direction also might have influenced the shape of the personal space (*Rios-Martinez, Spalanzani & Laugier, 2015*); therefore, considering the combinations of personal space and pre-touch reaction distance would be essential to realize more natural reactions.

The separation of touching the displayed agent or the GUI on a monitor is another issue for pre-touch interaction design. Recent smart kiosks (*e.g.*, "A.I. Smart Kiosk" by Advanced Robot Solutions) have a displayed agent and menu buttons on the same screen. In such situations, people move their hands toward the display to touch either one of the buttons or the displayed agent. Therefore, the system needs to distinguish between these two different options and recognize the touch intention of the people to achieve appropriate pre-touch reactions. For example, if a person wants to touch the displayed agent, a pre-touch reaction is natural, although such a reaction should be suppressed toward button-touching behaviors.

One possible future work based on this study is to investigate the relationships between cultural differences and pre-touch reaction distances. Similar to the cultural differences in personal spaces reported in several studies (*Beaulieu, 2004*; *Sorokowska et al., 2017*; *Sussman & Rosenfeld, 1982*), pre-touch reaction distances might have been influenced due to the cultures.

### Limitations

This study has several limitations. First, the experiment in this study has a simple setting where participants only touched or moved their hands toward a display agent that had a human-like appearance. More complex situations, such as conversational settings or long-term interactions and different types of displayed agents (*e.g.*, not human-like appearances such as pets) might change the effectiveness of the pre-touch reactions toward the

participants' perceived feelings. Moreover, this study investigated the basic perceived feelings toward the displayed agents. To investigate the additional impressions related to the user experiences, measuring the social acceptance scale (*Heerink et al., 2008*) would be useful while the displayed agents provides services in real settings.

The developed system only recognized the touched events and the distance between the displayed agents' faces and the participant's hands. The effects of the characteristics of touch behavior have not yet been addressed. For example, a past study reported that the correlation between hand-approaching speed and pre-touch reaction distance (*Shiomi et al., 2018*), *i.e.*, people's pre-touch reaction distance, increases toward fast hand movements. People respond differently to aggressive touches and soft pats. To recognize such different kinds of touch behaviors, the developed system requires additional sensors, such as acceleration sensors on the display. An interesting future work is investigating different reaction behaviors depending on the touch characteristics.

## CONCLUSIONS

The aim of this study is to investigate the effectiveness of pre-touch reactions for displayed agents. For this purpose, a displayed agent system that detects touch events on a display and pre-touch behaviors around it is developed. This study experimentally investigated whether participants preferred pre-touch reactions while they were interacting with displayed agents by investigating perceived impressions. The experiment results showed that they significantly preferred pre-touch reactions over post-touch reactions in the context of the perceived impressions as anthropomorphism, likability, human likeness, naturalness, and total feelings.

### Funding

This work was supported by the Mitsubishi Electric Corporation and JST CREST (Core Research for Evolutional Science and Technology) Grant, Number JPMJCR18A1. There was no additional external funding received for this study. The funders had no role in study design, data collection and analysis, decision to publish, or preparation of the manuscript.

### Grant Disclosures

The following grant information was disclosed by the authors:
Mitsubishi Electric Corporation.
JST CREST (Core Research for Evolutional Science and Technology): JPMJCR18A1.

### Competing Interests

The author declares that they have no competing interests.

## Author Contributions

- Masahiro Shiomi conceived and designed the experiments, performed the experiments, analyzed the data, performed the computation work, prepared figures and/or tables, authored or reviewed drafts of the article, and approved the final draft.

## Ethics

The following information was supplied relating to ethical approvals (*i.e.*, approving body and any reference numbers):

The Advanced Telecommunications Research Institute International Review Board Ethics Committee approved the study (501-4).

## Data Availability

The raw data are available as a Supplemental File.

## Supplemental Information

Supplemental information for this article can be found online at http://dx.doi.org/10.7717/peerj-cs.2277#supplemental-information.

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
