# Peer review of "Pre-touch reaction is preferred over post-touch reaction in interaction with displayed agent"

_PeerJ Computer Science, doi:10.7717/peerj-cs.2277_

## Round 0.1 · original submission · Major Revisions

The reviewers agree on the interestingness of the proposed topic and solutions and on the general quality of the presented work.

However, there are some issues that need to be solved before considering the paper for publication.

Please, see that a major revision is suggested to allow the reviewers to check out some major concerns on your work.

You must clarify the concept of “distance” and provide a clear assessment of the related factors as suggested by multiple reviewers. Please, improve the clarity of the experiment description to improve the experiment reproducibility.

Please, provide more details on the participants and a comment related to the possible limitations of the study in terms of participant recruitment (e.g., nationality, age range, and so on). Moreover, a clear description of the terminology is required to allow a better understanding of the topic and results. Provide also a description and explanation for all the used symbols and abbreviations. Please, revise attentively the section dedicated to the results.

Consider the reviewers’ suggestions concerning the organisation of the paper and answer all the reviewers’ concerns and suggestions with a point-by-point response.

Additionally, I understand that you have contributed to the field with many works on this topic, for which I am very grateful, but I suggest you consider decreasing the number of self-citations (which exceeds the 15% of the total citations) reporting only the ones that are necessary to deliver your novel work. See that this is just a suggestion and there is no obligation in taking some actions about it.

Reviewer 1 ·

Basic reporting

The article aims at investigating how different types of human-machine interaction methodologies (pre-touch or post-touch reactions) affect the user's perception of an avatar in the context of display-based (2D) agents.
The article is well structured and the research question is clear. Despite that, several sections result to be poorly written [1-2], confusing [3-4] or trivial [5] and could be simplified.
Moreover, the authors did not mention how they dealt with the issue presented at line 47 [6]. It results unclear whether they decided to consider the distance between user hand and avatar face or between user hand and display physical boundary. In fact, at line 144, the authors argue that the UltraLeap sensor "calculates the distance between the display surfaces [...] and the detected hands of the participants." Thus implying that the distance measured is between user hand and display.
However, in Figure 3, it looks like the only difference between the 20cm condition and the 45cm condition is the distance of the agent within the virtual environment, while the user hand is at the same distance with respect to the screen.
For a better understanding and reproducibility of the experiment, it should be clarified:
a) which distance was actually measured,
b) what is the rationale that motivated the authors to measure one distance over the other.

Moreover, the objective of the "Implications" section should be stated more clearly, as it currently enumerates a series of state-of-the-art issues (e.g. define a logic to manage pre/post touch behavior according to the position of the agent [7], pre-touch reactions in teleoperated avatars [8], etc.) that are not addressed in the research and that would better fit the "Related works" section.

[1] (line 20) "To design the pre-touch reaction of such an agent, we focused on the display's physical boundary as a criterion for pre-touch by agents on the display."
[2] (line 89) "For example, Shiomi et al. modeled the pre-touch reaction distance around the face from a data collection with people and implemented a model to investigate the importance of pre-touch reaction distance toward a social robot (Shiomi et al. 2018b)."
[3] (line 78) "Moreover, past studies have focused on spatial interaction with such displayed agents, e.g., a scene where people encountered the displayed agent, and reported how their interaction distances differed based on the people's impressions toward the displayed agents (Cafaro et al. 2016)."
[4] (line 278) "The teleoperated avatar's pre-touch reaction resembles a visualization of remote touch interaction in distant locations."
[5] (line 46) "One notable difference between the displayed agents and robots/agents is the existence of a display."
[6] (line 47) "Unlike robots in physical environments and agents in VR environments, it is unclear whether the criterion for the pre-touch reaction distance to which the display agents should respond is the distance from the face on the displayed image or to the display's physical boundary."
[7] (line 268) "Modifying the reaction behaviors, depending on the places that are actually touched would be useful for refusing excessive or 'bad' touches, as suggested by a past study (Kimoto & Shiomi 2024)."
[8] (line 277) "From another perspective, a pre-touch reaction is useful for a teleoperated avatar to increase the perceptions of being touched by the operator."

Experimental design

The experiment is overall well designed, it comprises a within-subject study with 4 different conditions that are ordered randomly to avoid repetition bias.

Nevertheless, the testing of the application is quite trivial [1], therefor a larger sample might have been useful to achieve a higher power test.
Was a Power Analysis carried out to evaluate the necessary sample size?

Moreover, a more sound motivation for the choice of the questionnaire could be appreciated.
Are there any validated questionnaire that could have been used to evaluate the user experience?

Eventually, a slightly more thorough description of the participants could be useful.
Did they participate on a voluntary basis? Were they students, academics, strangers, etc. ? Also, the Standard Deviation of the age could be an interesting parameter to better understand the sample.

[1] (line 294) "[...] we conducted our experiment in a simple setting where participants only touched or moved their hands toward a display agent that has a human-like appearance."

Validity of the findings

no comment

·

Basic reporting

The article proposes a novel approach to enhance user experience when interacting with displayed autonomous agents, specifically by focusing on "pre-touch" interaction. This interaction technique enables the agent to respond just before the user touches the screen, thereby improving the overall interaction experience.

The authors deliver a thorough introduction to the problem and offer a comprehensive literature review. Their explanation is accessible, even to those not well-versed in the field. The English is correct and fluent, and the paper's structure is well-organized and appropriate.

Experimental design

To validate their approach, the authors employ an experimental design derived from the literature. The experiments and the sample are appropriately described.

One minor concern is the nationality of the participants, as all of them are Japanese. This could potentially influence the results. I understand that it would be almost impossible to represent various nationality in this kind of experiment; hence I believe the authors' selection is acceptable. However, it is important to consider that this factor might influence the outcomes. If there are studies in the literature that analyze the impact of participant nationality on such experiments, it would be beneficial to cite them and briefly mention this issue.

Validity of the findings

Considering the experimental design and sample size, I find the results to be valid, though not conclusive, which is expected for this type of experiment.

One concern I have is about the description of the results. The authors use concepts such as "anthropomorphism," "likeability," "human likeness," "naturalness," and "total feeling." Although the articles defining these concepts are properly cited, I believe the paper should include at least a brief description of each. Additionally, the "Results" section contains many symbols that are never described. Even if their meanings can be inferred, it is essential to explain all symbols before using them. Moreover, readers from the computer science community might not be familiar with this type of analysis and may not understand the symbols if they are not properly introduced.

·

Basic reporting

The work fits well in the field of the study. The analysis proposed is introduced by detailed literature references that explore the field background and highlight the novelty of this work with reference to the state of art. There are some minor typos in the reference format used. According to the Guideline, the references must be defined using the 'Name. Year' style (e.g. Smith et al., 2005), thus with a comma after the name of the authors and before the year of publication. In many of your citing you have missed the comma between the two parts.

In general, the figures are of good quality and help the reader to understand the content of the text. In my opinion, some minor issues should be considered concerning Figure 2 and Figure 3. In Figure 2, the text is easy to read but the direction of the arrows is not clearly visible (e.g. the arrow exiting the screen and entering the “Touchscreen” block). I suggest the authors slightly increase the size of the arrows to make them clearer. With reference to Figure 3, instead, the picture depicting the pre- and post-touch reactions of the displayed agent are small and difficult to compare without zooming. Again, I suggest the authors increase the size of the pictures if possible. Finally, I think there may be a typo on line 183 when figure 3 is cited. The text reports that the left column of the image shows the “pre-touch level reaction”, but this contrasts with the title of the left column (“Post touch interaction”).
As a minor typo, according to the Guideline, on line 182 I suggest that the authors refer to Figure 2 using “Figure” instead of "Fig”.

The supplemental material is well anonymised and reports the data collected during the experiment. However, it should be better detailed to simplify the reader's understanding and the reproducibility of results. In particular, the meaning of columns header in "Pre-touch_Data.xlsx" document should be better explained by reporting, for example, the meaning of the items identified as "PF", "PB", "TF", and "TB".

The content of the manuscript is well-detailed and described. The subsections that divide the body of work help the reader to focus their attention on the main parts of the experimental procedure, making it easier to understand the content of the text. However, I think that some subsections are too small or redundant and can therefore be removed or integrated into other existing parts of the document. For instance, the “Task design” subsection of the “Materials & Methods” section is very short and is only used to remark on the topic of the analysis. Therefore, it may be incorporated into an existing subsection, such as the "Hypothesis and prediction" one.

Experimental design

The experimental protocol is described with sufficient details and information to replicate them. If there is a weakness, it is in the description of distance factor. In some cases, I think that this term (distance) is used with two different meaning inside the text that make sometimes difficult to understand exactly which of the two meanings is being referred to. For example, in section “Condition” the term “distance” is considered with the reference to the viewpoint on the displayed agents, therefore it is about the perceived distance between the participant and the displayed agent. On the other hand, in line 187 the term Distance is defined as “20 cm from the display surface/45 cm from the display surface” that makes them similar to what you define as pre-touch reaction distance. Please, better details this point to make them clearer.

The results of the questionary analysis are reported in term of statistical measures but lack of information about the type of data that have been compared using the ANOVA test. In “Condition” section only two factors are reported: the touch and the distance. However, in the “Results” section are also reported comparison with reference to another factor: the interaction effect. How do you have evaluated this other aspect? Is it possible to evaluate this other factor starting from the results depicted in Figure 4? Another information that is not reported in the main text concern the values of “anthropomorphism” and “likeability” used in the comparison. Both the aspects are described in your work with a single representative value. However, during the experiment, you use the multi-item Godspeed scale questionary to collect information about them. How did you evaluate this single value starting from the different items? Please, if possible better details this aspect.

Validity of the findings

no comment

---

## Round 0.2 · Minor Revisions

Thank you for addressing all the major concerns raised by the reviewers.

The manuscript is almost ready for publication but requires solving just two additional comments regarding a typo and the lack of clarity in respect to how the authors refer to the study.

I think the reviewer would like to point out that sometimes referring to the study in an active way or instead of the researchers is somewhat confusing.

For example, in line 136 you state “This study installed a depth sensor [..]”, while it would be clearer if you stated something like “In this study, a depth sensor is installed [..]”.

I hope these suggestions can help you in addressing this final issues.

Reviewer 1 ·

Basic reporting

The authors have made the requested changes.

Experimental design

The authors clarified doubts related to power analysis

Validity of the findings

no comment

Additional comments

no comment

·

Basic reporting

The authors have successfully addressed my main concerns from the previous version. They have explained the limitations and considerations regarding the nationality of the participants, all of whom were Japanese. Additionally, I suggested clarifying some terms and abbreviations that might be unfamiliar to a computer science audience. This issue has also been effectively resolved. Therefore, I believe the paper is now ready for publication.

Experimental design

The experimental design was already properly described in the previous version.

Validity of the findings

The findings are valid, although not conclusive.

·

Basic reporting

The changes made to the manuscript solved almost all the problems I had pointed out in the previous review. Here a couple of comments about the new version:

1) I think there is a typo in line 224, where the word “this” is written in capital letter despite being after the comma.

2) In my opinion the use of “This study” instead of “We” is sometimes a bit confusing, because it seems like “the study” is able to make decisions and develop systems. For example, in line 150 it appears that the study was able to install sensors, or in line 239 seems like the study itself prepared the questionary items. If possible, I suggest the author modify these parts.

Experimental design

no comment

Validity of the findings

no comment

---

## Round 0.3 · accepted · Accept

Thank you for addressing all the reviewers’ concerns.

The reviewers were already satisfied by the content, but there was just a minor issue with the delivery of some parts of the manuscript. Therefore, the reviewers have been not involved in this final revision phase.

The manuscript is now ready for publication.